# Quality Detection and Grading of Rose Tea Based on a Lightweight Model

**DOI:** 10.3390/foods13081179

**Published:** 2024-04-12

**Authors:** Zezhong Ding, Zhiwei Chen, Zhiyong Gui, Mengqi Guo, Xuesong Zhu, Bin Hu, Chunwang Dong

**Affiliations:** 1College of Mechanical and Electronic Engineering, Shihezi University, Shihezi 832000, China; dingzezhong1203@163.com; 2Tea Research Institute, Shandong Academy of Agricultural Sciences, Jinan 250100, China; zv.chen@foxmail.com (Z.C.); zhiyonggui_zstu@163.com (Z.G.); qq2524434302@163.com (M.G.); 3Feilanda Intelligent Technology Co., Ltd., Hangzhou 311121, China; 15831111103@163.com

**Keywords:** rose tea grading, YOLOv8, attention mechanism, lightweight, SIoU

## Abstract

Rose tea is a type of flower tea in China’s reprocessed tea category, which is divided into seven grades, including super flower, primary flower, flower bud, flower heart, yellow flower, scattered flower, and waste flower. Grading rose tea into distinct quality levels is a practice that is essential to boosting their competitive advantage. Manual grading is inefficient. We provide a lightweight model to advance rose tea grading automation. Firstly, four kinds of attention mechanisms were introduced into the backbone and compared. According to the experimental results, the Convolutional Block Attention Module (CBAM) was chosen in the end due to its ultimate capacity to enhance the overall detection performance of the model. Second, the lightweight module C2fGhost was utilized to change the original C2f module in the neck to lighten the network while maintaining detection performance. Finally, we used the SIoU loss in place of the CIoU loss to improve the boundary regression performance of the model. The results showed that the mAP, precision (P), recall (R), FPS, GFLOPs, and Params values of the proposed model were 86.16%, 89.77%, 83.01%, 166.58, 7.978, and 2.746 M, respectively. Compared with the original model, the mAP, P, and R values increased by 0.67%, 0.73%, and 0.64%, the GFLOPs and Params decreased by 0.88 and 0.411 M, respectively, and the speed was comparable. The model proposed in this study also performed better than other advanced detection models. It provides theoretical research and technical support for the intelligent grading of roses.

## 1. Introduction

In the last several years, the commercial value of roses has become increasingly important [1]. Roses can be used in perfume, rose tea, and other applications [2]. Currently, the rose planting area in Pingyin County, Jinan City, Shandong Province is 61,000 acres. Annual production of 3000 tons of processed dried rose flowers. Dried roses are mainly used to make rose tea, but rose tea without grading is not competitive in the market. Grading rose tea not only broadens the price range but also facilitates consumer purchases [3]. At present, rose grading still requires manual labor, which is time-consuming and inefficient. Moreover, the manual grading of one kilogram of rose tea will increase the cost by 10 yuan. Therefore, there is an urgent need for mechanical grading to replace manual grading. However, applying a model to actual production lines may produce problems such as poor hardware performance [4], and the current models have a large number of parameters and high computational complexity, which is not conducive to deployment. Thus, it is necessary to design a high-precision and lightweight rose tea quality detection and grading model.

Deep learning technology has made rapid progress in agriculture in the last several years [5,6]. At present, many scholars have conducted a lot of research on detection and grading in agriculture [7]. Du et al. put forward a DSW-YOLO model to accurately detect ripe strawberries and their occlusion levels. Their model achieved excellent detection accuracy [8]. Liu et al. proposed an efficient channel pruning method based on YOLOX for the detection and grading of shiitake mushrooms. Their method could effectively detect and grade shiitake mushrooms [9]. Li et al. put forward a lightweight tea bud detection model based on the improved YOLOv4. Their model detected tea buds with an accuracy of 85.15%, which was 1.08% greater than the average accuracy of the original model, and the number of parameters decreased by 82.36% [10]. In addition to the above research, studies have also been done in the field of flower detection and grading. Cıbuk et al. proposed a deep convolutional neural network (DCNN)-based hybrid method that was applied to the classification of flower species. It used a pre-trained DCNN model for feature extraction and an SVM classifier with a radial basis function kernel to classify the extracted features with high classification accuracy [11]. Tian et al. proposed a deep learning method using the YOLOv5 algorithm to achieve the fine-grained image classification of flowers. It was able to successfully identify five different types of flowers [12]. Zeng et al. proposed a new lightweight neural network model based on multi-scale characteristic fusion and attention mechanisms. Their model had fewer parameters and high classification accuracy [13]. Wu et al. proposed a real-time apple flower detection method using the channel-pruned YOLOv4 deep learning model, and the model was pruned using the channel pruning algorithm, which achieved fast and accurate detection of apple flowers [14]. Shang et al. proposed a lightweight YOLOv5s model for apple flower detection by replacing the original backbone with ShuffleNetv2 and replacing the Conv module in the neck part with the Ghost module [15]. Li et al. detected and identified kiwifruit flowers using YOLOv5l. They classified kiwifruit flowers into ten categories and clusters and branch knots into four categories. The mAP for all-species detection was 91.60%, and the mAP for multi-class flowers was 93.23%. It was 5.70% higher than the other four categories. It has high accuracy and speed for detection and classification [16].

The above research was conducted to detect common objects and classify common flowers. Previous studies mainly focused on the grading of different types, but this study focuses on the same type of flower. The detection and grading of kiwifruit flowers are similar to the work carried out in this study. However, kiwifruit flowers are detected and graded outdoors, while rose tea is detected and graded indoors. Moreover, the difference between the two flowers is significant. This algorithm is not suitable for detecting rose tea. Currently, there are few reports on the detection and grading of rose tea within the class. Rose tea is similar in color, and some flowers are similar in shape, which makes their detection and grading more difficult. At the same time, future applications in actual production may involve problems related to poor hardware performance. Therefore, this study proposes a lightweight rose detection and grading model based on the improved YOLOv8.

The main contributions of this paper are as follows: (1) Four attention mechanisms are respectively added to the backbone of the experiments. We compare the experimental results and choose the CBAM to enhance the detection performance of the model. (2) The C2f module is substituted by the module C2fGhost in the neck of the network to achieve lightweighting while maintaining performance. (3) In terms of the loss function, the original CIoU loss is substituted by the SIoU loss to improve the boundary regression performance of the model.

## 2. Methods

### 2.1. The Abbreviations Used in This Article and the Experimental Design Flowchart

The abbreviations used in this article and the experimental design flowchart are shown in Table 1 and Figure 1, respectively.

### 2.2. The YOLOv8 Network

YOLOv8 is the latest YOLO model for object detection, instance segmentation, and image classification, and it offers new features based on previous YOLO versions to improve performance and flexibility. According to the ratio of network depth and width, YOLOv8 can be categorized into five types: n, s, m, l, and x. Given the model size and complexity, YOLOv8n was chosen as the base network model for this study. The four components of YOLOv8n are the input, backbone, neck, and head [17], as shown in Figure 2a.

The model input is augmented with mosaic data, and an anchor-free mechanism is used to directly predict the center of the object, which reduces the number of anchor frame predictions and accelerates the non-maximal suppression. The function of the backbone is to extract the information featured in the picture. YOLOv8n’s backbone references the structure of CSPDarkNet-53 and uses C2f instead of the C3 module. The gradient flow is increased, the level of computation is significantly reduced, and the convergence speed and convergence effect are significantly improved. The neck fuses the features between the backbone and the head. The neck takes advantage of the PANet structure, which unifies the network’s top and lower information flows and improves detection capabilities. Using the features that were extracted, the head makes predictions. YOLOv8n’s head is a decoupled head like YOLOX, and it has three output branches. Each output branch is subdivided into a regression branch with a DFL strategy and a prediction branch [18,19].

### 2.3. YOLOv8n Network Improvements

In this study, the YOLOv8n network was improved. A diagram of the network after the improvement is shown in Figure 2b. First, four types of attention mechanisms are introduced into the backbone of the network for the experiments. To achieve lightweighting without sacrificing detection performance, the original C2f module of the network is replaced in the neck by the lightweight C2fGhost module. Finally, the model’s boundary regression performance is enhanced by replacing the CIoU loss with the SIoU loss.

#### 2.3.1. The Attention Mechanism Module

In object detection algorithms, the purpose of the attention mechanism is to apply more weight to the information to help solve a problem in a specific scenario by ignoring the irrelevant information and focusing on the key information, thereby improving detection performance. In this study, we chose four attention mechanisms, the CBAM [20], CA module [21], ECA module [22], and NAM [23], with which to conduct experiments.

##### The Convolutional Block Attention Module

The CBAM is a lightweight attention module that combines channel and spatial attention mechanisms along two independent dimensions, as shown in Figure 3. Channel attention aggregates the spatial information related to features through average pooling and maximum pooling, compresses the spatial dimensions of the features, and feeds them into a shared network that adaptively adjusts its weights through learning to generate attention weights. Spatial attention, on the other hand, executes maximum pooling and average pooling per channel and then pools all channels for the same feature point. The feature maps are superimposed to generate spatial attention weights. The optimized feature map is ultimately produced after the feature maps first go through the channel attention module, where they receive the channel attention weights and multiply them by the initial features. Next, they enter the spatial attention module, where they receive the spatial attention weights and multiply them by the features from the previous step.

##### The Coordinate Attention Module

Figure 4 illustrates the two primary phases of the CA module. Varying channels are given varying attention weights by the CA module, which is a fundamental channel attention mechanism. It typically employs global average pooling and a fully connected layer to learn the degree of correlation between channels, and then it applies a softmax function to normalize the attention weights.

##### The Efficient Channel Attention Module

The ECA module, as shown in Figure 5, adopts a 1 × 1 convolutional layer directly after the global average pooling layer. It removes the fully connected layer, which makes dimensionality reduction unnecessary and captures cross-channel interactions efficiently. ECANet requires a few parameters to produce good results. ECANet uses one-dimensional convolution to achieve cross-channel information interaction. To increase the frequency of cross-channel interaction for layers with several channels, the convolution kernel’s size is adaptively changed.

##### The Normalization-Based Attention Module

As seen in Figure 6, the NAM is a compact and effective attention mechanism that redesigned the channel attention and spatial attention sub-modules while adopting the CBAM’s module integration. In the channel attention sub-module, the scaling factor is used in batch normalization. The scaling factor shows the significance of each channel as well as the amount that it has changed.

#### 2.3.2. The C2fGhost Lightweight Module

GhostNet is a lightweight network that was designed by Huawei’s Noah’s Ark Lab in 2020. The GhostNet lightweight network model can maintain the size and channel size of the original convolutional output feature map while reducing the computational and parameter requirements of the network. First, a small number of ordinary convolution kernels are employed to take feature data out of the input feature map. Then, linear transformation operations are performed on the feature map, which is less computationally expensive than ordinary convolutions. Finally, the final feature map is generated through concatenation, as shown in Figure 7a. It increases feature expressiveness by introducing additional branches into the convolution operation. The lightweight module C2fGhost replaces the bottleneck in the C2f module of the original network with Ghost BottleNeck, as shown in Figure 7b. It makes use of the truncated gradient flow technique and the cross-stage feature fusion strategy to increase the network’s learning capacity, lessen the impact of redundant information, and improve the variability of learned features across various network levels. The introduction of the C2fGhost module greatly reduces the number of model parameters needed as well as the computational effort by greatly reducing the number of common 3 × 3 convolutions [24,25].

#### 2.3.3. Loss Function

The YOLOv8 algorithm adopts DFL loss + CIoU loss as the regression loss. There is some ambiguity surrounding CIoU in terms of the relative values described by the aspect ratio. In this study, SIoU loss is used in place of CIoU loss.

SIoU loss is a function that takes into account the angle of the predicted regressions and redefines the metric for the angle penalty. It allows the frame to drift to the nearest coordinate and then return to one of the coordinates. This method can reduce the total degrees of freedom. It is composed of four parts: the angle cost, the distance cost, the shape cost, and the IoU cost [26]. Its calculation schematic is shown in Figure 8.

The calculation is as follows:(1)Λ=1−2sin2⁡arcsinchσ−Π4Δ=1−ⅇ−(2−Λ)bcxgt−bcxcw+1−ⅇ−(2−Λ)bcygt−bcychΩ=1−ⅇ−w−wgtmax⁡w,wgtθ+1−ⅇ−h−hgtmax⁡h,hgtθLIoUcost=1−IoU=1−IntersectionUnion
where *Λ* represents the angle cost, Δ represents the distance cost, *Ω* represents the shape cost, and *L_IoUcost_* represents the IoU cost.

Finally, the SIoU loss calculation is shown in Equation (2):(2)$$L_{box}=1-IoU+\frac{\Delta +\Omega }{2}$$

## *3.* Experimental Design and Result Analysis

### 3.1. Dataset Production

This experiment uses rose tea from Pingyin (116.45° E, 36.28° N), Shandong Province, as the research object. The rose tea was divided into seven grades, which included super flower, primary flower, flower bud, flower heart, yellow flower, scattered flower, and waste flower. A super flower should not be yellow or white in color, and the heart of the flower should not be exposed. The standard for the primary flower is to have a little bit of the flower heart, and the color must not be turning white or yellow. All the flowers must be in full bloom. The standard for the flower bud is that the surrounding petals cannot be blooming or turning yellow. The standard for the flower heart is that the color of the flower is good, it cannot turn white, cannot be too small, and half or all of the flower hearts are exposed. Yellow flower refers to the yellowing of the entire flower or more than half of it, with a large yellow heart; a scattered flower is a flower larger than the bud with a middle that has a hard heart, the surrounding petals are scattered, and the color cannot be turning yellow. Waste flowers are the flowers that are left over after the other six types of flowers have been selected, and compared to the other six types of flowers, the waste flowers are broken, moldy, and of poorer quality. Figure 9a shows different grades of rose tea, excluding the waste flower. Because the waste flower contains more kinds of flowers, it is the one remaining after the selection of the six kinds of flowers; they are crushed, moldy, and of poor quality. There is no value to the selection, so the image of the waste flower is not shown.

We created a dataset of 1500 images taken using a Canon camera (Canon EOS80D) and a cell phone. We used two cameras to take pictures with the purpose of getting images at different resolutions and enriching the dataset. The dataset was first randomly sorted into a training set + validation set and a test set according to the ratio of 4:1. The training set and validation set were then divided according to the ratio of 4:1, and LabelImg (Tzutalin, US) was used to annotate them and generate label files. The dataset contained more than 9000 ground-truth boxes. Figure 9b shows the total number of ground-truth boxes for each category.

### 3.2. Experimental Environment and Parameter Settings

The operating system used for this experiment was Windows 10, the CPU model was a 13th-Gen Intel Core i7-13700F, and the GPU model was a NVIDIA GeForce RTX4070. The programming language was Python 3.9, the deep learning framework was PyTorch 1.8.2, and the GPU acceleration library was CUDA 11.1 and CUDNN 8.4.1. All the experiments in this study were carried out on the PyTorch deep learning framework, using the Adam optimizer to update the parameters. A total of 200 epochs were trained, the batch size was 8, and the momentum was set to 0.937.

### 3.3. Indicators of Model Evaluation

The experiments adopted common evaluation metrics for object detection tasks to assess the performance of the experimental results. These evaluation metrics include: precision (P), recall (R), mAP, Params, GFLOPs, and FPS [27].

(1) Precision is the proportion of correct positive predictions to the proportion of all positive predictions, which is calculated as Equation (3):(3)$$P=\frac{TP}{TP+FP}$$
where *TP* represents true positive, which means the number of actual positive examples predicted as positive examples, and *FP* represents false positive, which means the number of actual negative examples predicted as positive examples.

(2) Recall is the proportion of positive cases in the sample that are predicted correctly, and this is calculated as in Equation (4):(4)$$R=\frac{TP}{TP+FN}$$
where *FN* represents false negative, which means the number of actual positive examples predicted as negative examples.

(3) mAP is the average of the detection accuracy of all categories and is calculated as in Equation (5):(5)$$mAP=\frac{1}{n}\sum_{i=1}^{n}AP_{i}$$
where *AP* represents the average precision of a single category.

(4) Params is the number of learnable parameters in the model, which reflects the complexity and resource consumption of the model.

(5) GFLOPs represents the number of floating point operations performed during the model inference process, which is related to the computational complexity of the model.

(6) FPS (frames per second) refers to the speed of the model.

### 3.4. Experimental Results and Analysis

#### 3.4.1. Attention Mechanism Comparison Experiment

Under the same conditions, each of the four attention mechanism modules was added to the backbone network for comparison. The experimental results are shown in Table 2.

After adding each of CA, ECA, NAM, and CBAM, respectively, to the backbone part of the original YOLOv8n model, the detection performance of the model was improved, and the speed of the model decreased, which suggests that the attention mechanism facilitated the network in extracting the key features of the rose tea, and it inevitably increased the model’s GFLOPs and Params. The best improvement was achieved with the CBAM, where the mAP, P, and R increased by 0.59%, 1.33%, and 0.99%, respectively.

#### 3.4.2. Performance of the Improved Model

The loss curve, log-average miss rate, P–R curve, and AP comparison with various flowers of the improved model after it was trained for 200 epochs are shown in Figure 10.

From Figure 10a, it can be seen that as the number of training epochs increases, the loss value of the improved model rapidly decreases and stabilizes. The final loss value converges near 2.08, and the improved model achieves good training results. The log-average error rate of the improved model for the classification of various types of flowers can be seen in Figure 10b. The log-average error rates for super flower, primary flower, flower bud, flower heart, yellow flower, scattered flower, and waste flower are 0.22, 0.39, 0.29, 0.22, 0.19, 0.43, and 0.35, respectively. As can be seen in Figure 10c,d, the detection AP values of the improved model for super flower, primary flower, flower bud, flower heart, yellow flower, scattered flower, and waste flower are 90.24%, 85.69%, 82.08%, 91.83%, 88.33%, 82.32%, and 82.61%, respectively. Compared with the original model, the AP values for super flower, primary flower, flower bud, scattered flower, and waste flower increased by 1.35%, 3.00%, 0.51%, 0.76%, and 1.22%, respectively. The mAP increased by 0.67%, and the overall performance of the model improved.

#### 3.4.3. Ablation Experiments

To verify the contribution of each module to the model proposed in this study, different modules were combined in the original model for the ablation experiments. The experimental results are shown in Table 3.

From Table 3, it can be seen that after the CBAM was added to the backbone part, the values of mAP, P, and R improved by 0.59%, 1.33%, and 0.99%, respectively, compared with the original model, with a slight decrease in speed. After the C2f module was replaced with the C2fGhost module in the neck part, the speed increased, and the mAP, P, and R values improved by 0.93%, 0.43%, and 0.47%, respectively, relative to the original model. The GFLOPs and Params values were reduced by 1.063 and 0.432 M, respectively. After replacing the loss function based on the improvements of the first two, the speed was comparable to that of the original model, and the values of mAP, P, and R improved by 0.67%, 0.73%, and 0.64%, respectively. The GFLOPs and Params values were reduced by 0.88 and 0.411 M, respectively, relative to the original model.

#### 3.4.4. Comparison of Model Effects before and after Improvement

As shown in Figure 11, three photos in the test set were randomly selected, processed, and then compared using the original YOLOv8n model and the improved model. As shown in the photos for Group 1, the improved model has higher confidence than the original model for detection and grading, and its detection and grading are more accurate. The original model mistakenly detects the flower bud as a yellow flower in the photos for Group 2. In the third group of photos, the original model has missed detections. However, the improved model avoids the problems of misdetection and missed detections of the original model, which further illustrates that the proposed model is more effective.

#### 3.4.5. Comparison between Different Object Detection Network Models

In order to further demonstrate the advantages of the improved model in this study in terms of detection performance and light weight, we compare the proposed model with the more advanced object detection models, including Faster R-CNN, SSD, YOLOv3, YOLOv4_Tiny, YOLOv5n, and YOLOv7, under the same conditions. The results of the comparison experiments are shown in Table 4.

From Table 4, it can be seen that the faster R-CNN model has a lower P value. The GFLOPs and Params are close to 50 times greater than the improved model, and the detection speed is slower than the improved model. The SSD model has a lower P value, a higher Params value, and a slower detection speed than the improved model. The YOLOv3 model has larger GFLOPs and Params values and lower P and R values than the improved model. The YOLOv4_Tiny model performs faster detection but has lower P and R values than the improved model. The YOLOv5n model has smaller GFLOPs and Params values than the improved model but slower detection speed and lower P and R values. Although the P and R values of the YOLOv7 model are higher than those of the improved model, the GFLOPs and Params values of the YOLOv7 are close to 13 times those of the improved model, and the detection speed is slower. The model in this study can meet the requirements for rose tea detection grading, although it achieves lower speeds than some models. Our comprehensive analysis indicates that the improved model in this study achieved the best performance of the more advanced models.

## 4. Discussion

Rose tea grades are uneven, making it uncompetitive in the market. While manual grading is time-consuming, labor-intensive, and inefficient, automated detection and grading is imminent. In addition, it may encounter problems such as low hardware performance in actual production, which makes it difficult to deploy. Therefore, this study proposes a lightweight rose tea quality detection and grading model based on an improved YOLOv8n network. Firstly, according to the experimental results, CBAM is selected from four different attention mechanisms. After adding CBAM, the mAP, P, and R of the model are improved by 0.59%, 1.33%, and 0.99%, respectively. The addition of the attention mechanism is beneficial to the extraction of the main features of the rose tea, but it also increases the GFLOPs and Params of the model. The original network C2f module is replaced by the lightweight module C2fGhost; the mAP, P, and R of the model are improved by 0.93%, 0.43%, and 0.47%, respectively, compared with the original model; and the GFLOPs and Params are reduced by 1.063 and 0.432 M, respectively. This model is lightweight and improves the model’s detection speed while maintaining accuracy. Finally, we replace CIoU loss with SIoU loss. mAP, P, R, FPS, GFLOPs, and Params values of the improved model are 86.16%, 89.77%, 83.01%, 166.58, 7.978, and 2.746 M, respectively. mAP, P, and R values are improved by 0.67%, 0.73%, and 0.64%, respectively, compared with the original model, and the GFLOPs and Params values were reduced by 0.88 and 0.411 M, respectively, with comparable detection speeds.

Although our improved model achieves lightweight performance while the detection performance is improved, it also obtains the best performance compared with the current advanced detection models. However, the model in this paper is only for rose tea in Pingyin County, Shandong Province, and it is necessary to further expand the data volume, improve the model generalization performance, and apply it to other regions for rose tea detection and grading. In addition, the deployment of the model in actual production will be a technical challenge, and it is necessary to design an effective deployment strategy to ensure that the model is successfully deployed to actual production as a means of promoting the development of the rose tea industry chain.

## 5. Conclusions and Future Research

In this paper, based on the YOLOv8n model, by adding the attention mechanism and replacing the lightweight structure and loss function, the established model achieved its lightweight status while meeting the requirements of rose tea detection and grading and providing technical support and theoretical research for the deployment of rose tea detection and grading and subsequent actual production of the model.

The detection and grading of rose tea is a novel research project. At present, the quality detection and grading of rose tea is still based on the appearance, shape, and color characteristics of this tea. On the basis of this research, in the future, we will combine the spectral image of rose tea to conduct a detailed analysis of the endoplasmic components of rose tea and finally combine the appearance, shape, and endoplasmic components of rose tea to achieve a more accurate and comprehensive detection and grading of rose tea.

## Figures and Tables

**Figure 1 foods-13-01179-f001:**
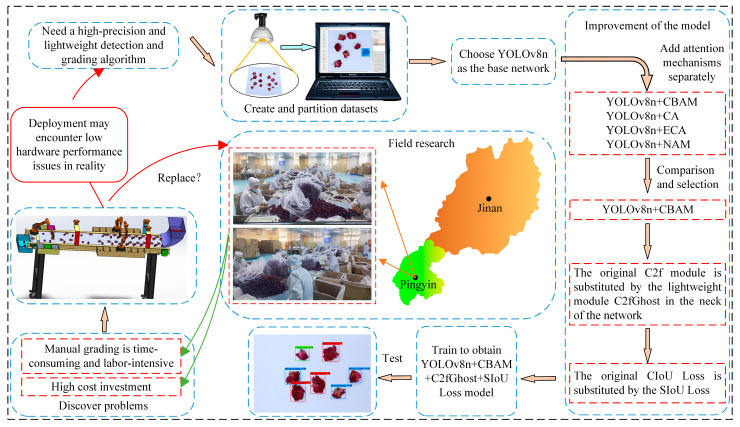
The analysis flowchart.

**Figure 2 foods-13-01179-f002:**
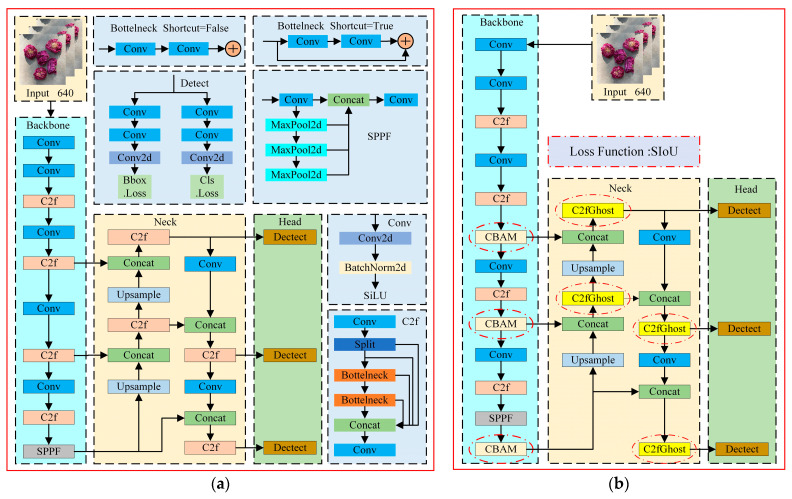
(**a**) Structure of the YOLOv8n model and (**b**) improved structure of the YOLOv8n model.

**Figure 3 foods-13-01179-f003:**
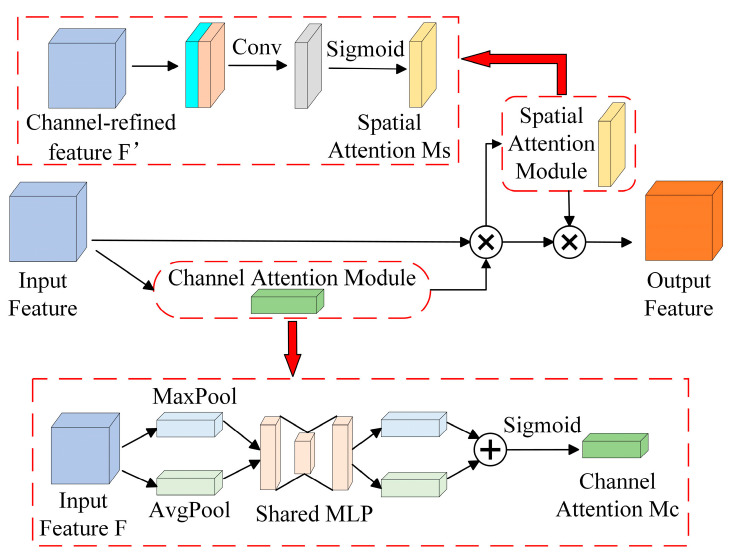
The convolutional block attention module.

**Figure 4 foods-13-01179-f004:**
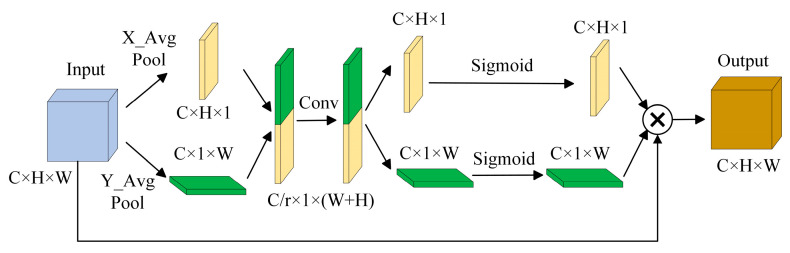
The coordinate attention module.

**Figure 5 foods-13-01179-f005:**
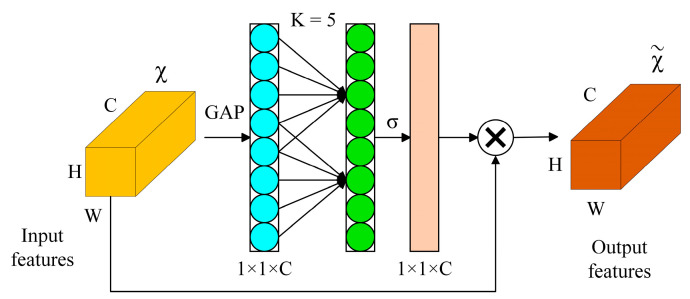
The efficient channel attention module.

**Figure 6 foods-13-01179-f006:**
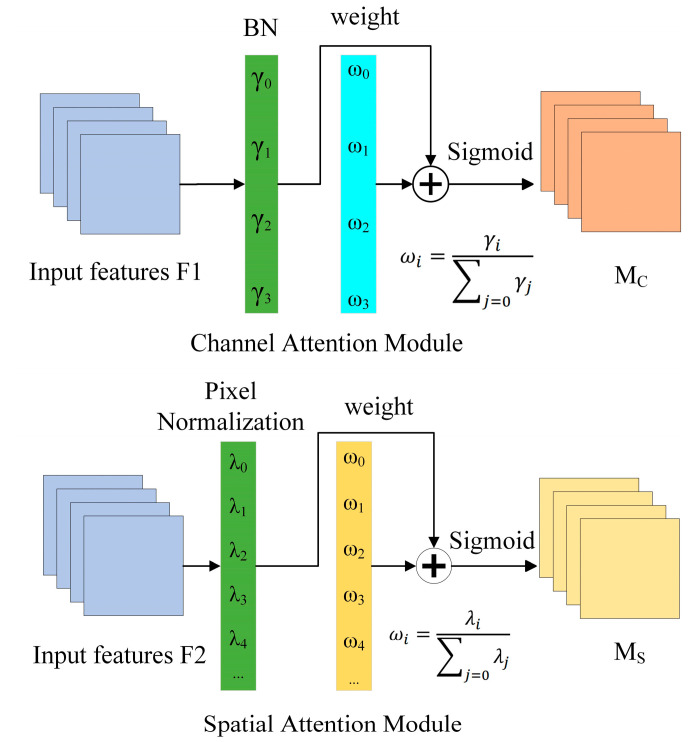
The normalization-based attention module.

**Figure 7 foods-13-01179-f007:**
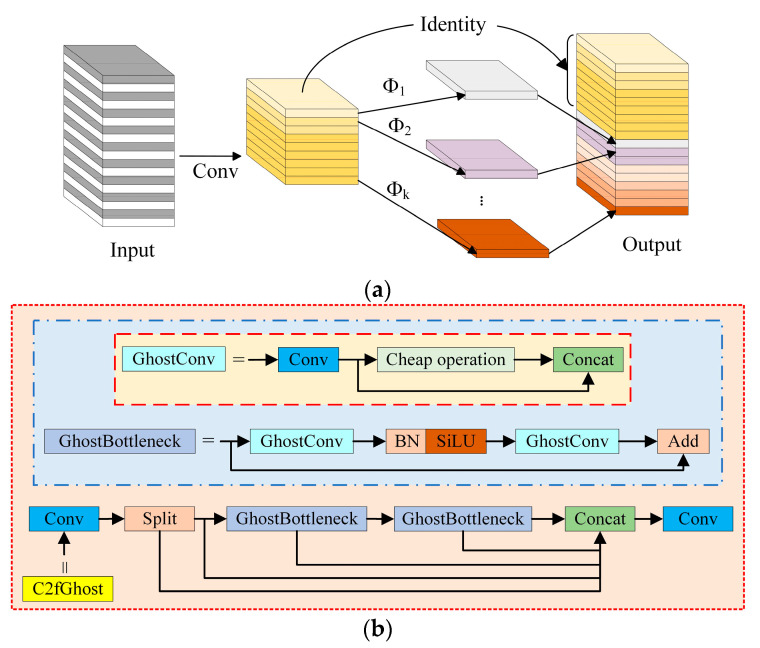
Lightweight module structure: (**a**) the Ghost module and (**b**) the C2fGhost module.

**Figure 8 foods-13-01179-f008:**
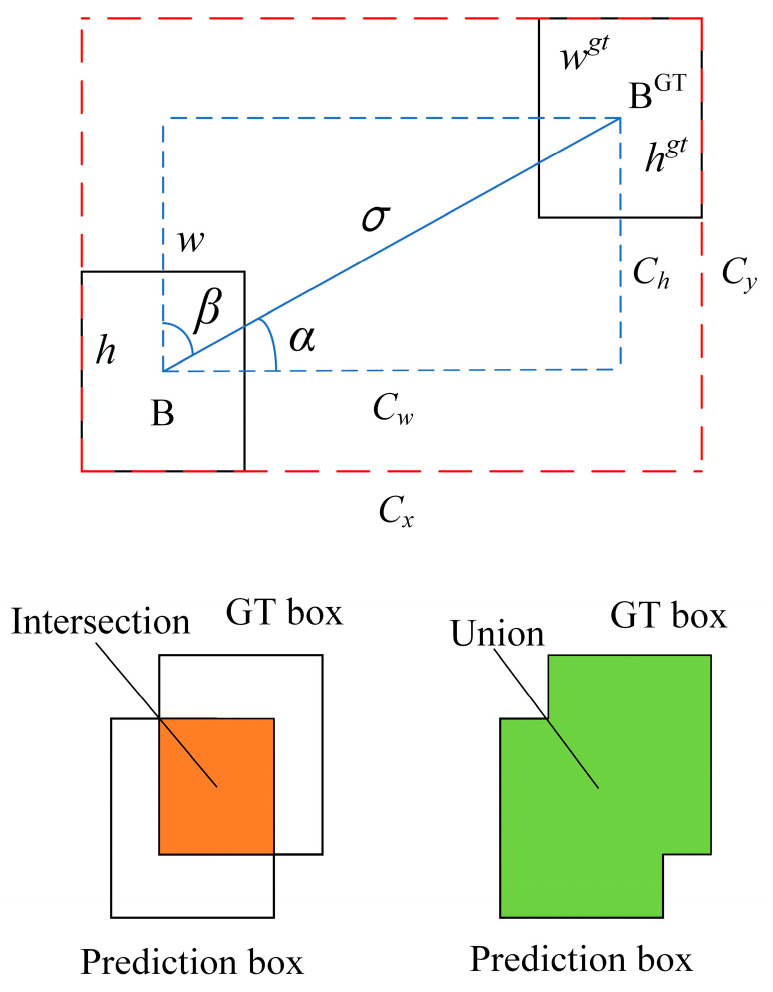
Schematic diagram for SIoU calculation.

**Figure 9 foods-13-01179-f009:**
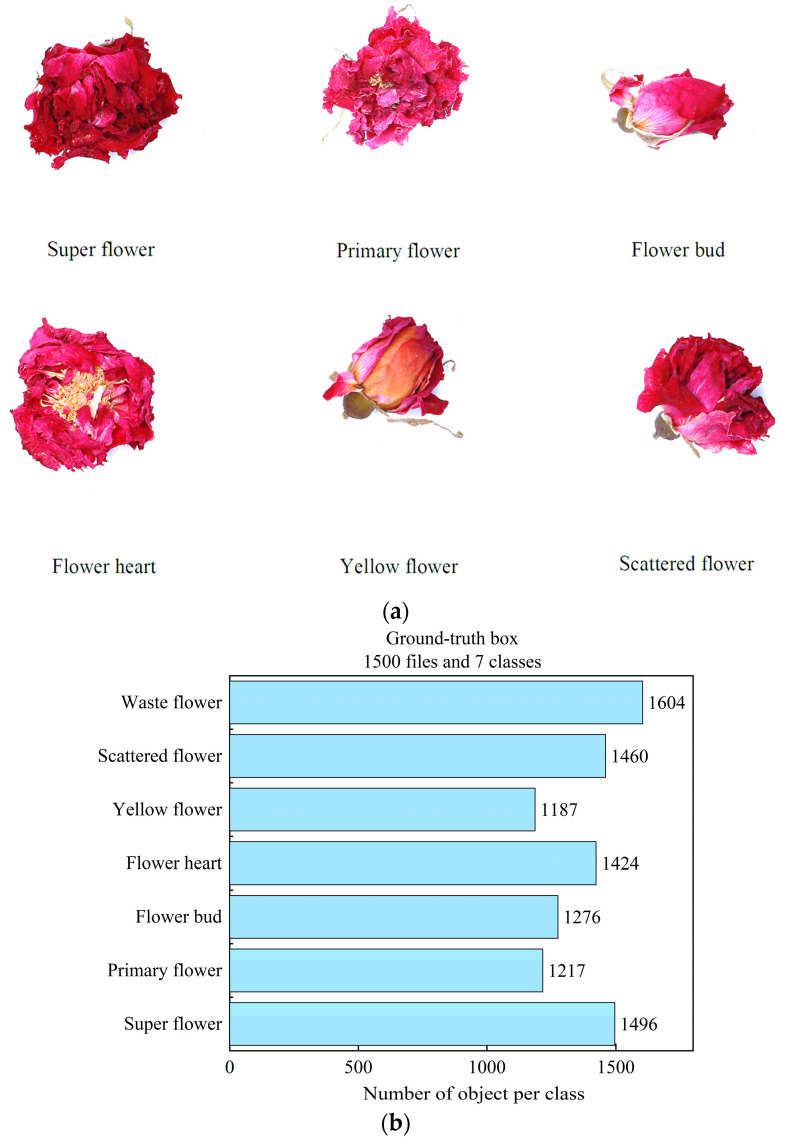
(**a**) Examples of roses of various grades and (**b**) the number of ground-truth boxes per category in the dataset.

**Figure 10 foods-13-01179-f010:**
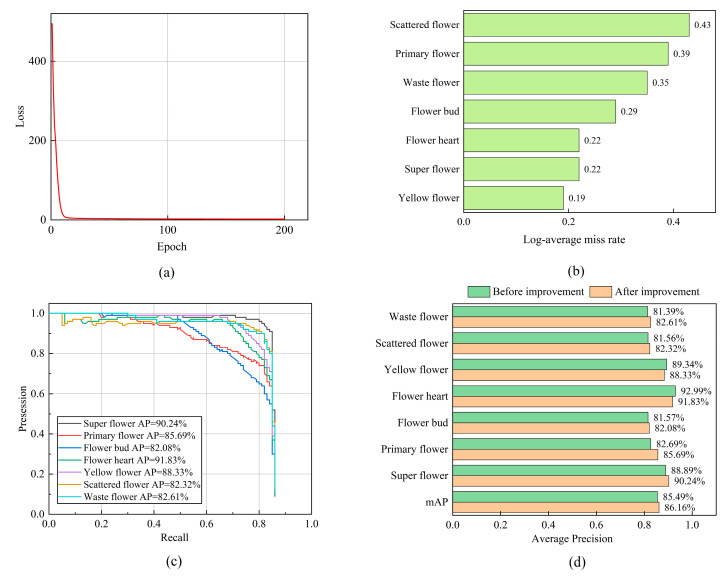
(**a**) Improved loss curve. (**b**) Improved log-average miss rate. (**c**) Improved P–R curve. (**d**) Comparison of various flower AP before and after improvement.

**Figure 11 foods-13-01179-f011:**
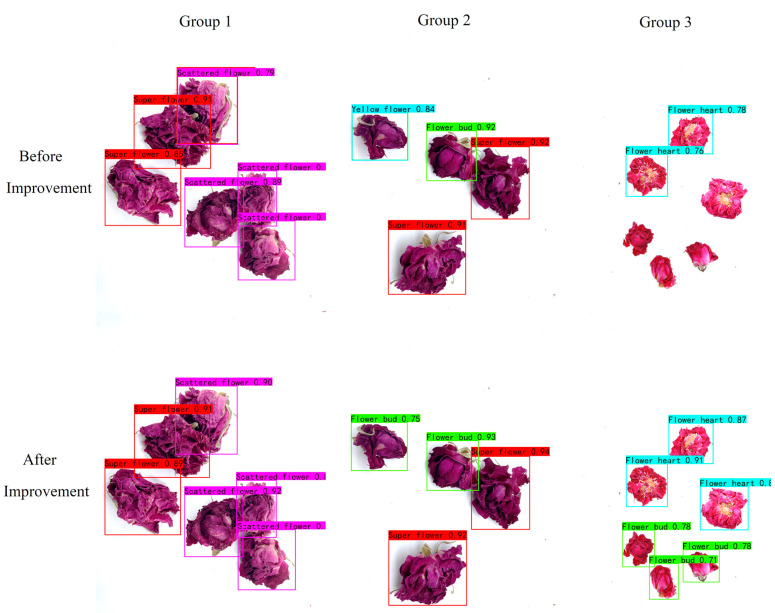
Comparison of detection results before and after improvement.

**Table 1 foods-13-01179-t001:** The abbreviations used in this article.

Number	Abbreviation	Full Name	Number	Abbreviation	Full Name
1	CBAM	Convolutional Block Attention Module	10	TP	True positive
2	CA module	Coordinate Attention Module	11	FP	False positive
3	ECA module	Efficient Channel Attention Module	12	FN	False negative
4	NAM	Normalization-Based Attention Module	13	TN	True Negative
5	AP	Average precision of a single category	14	P	Precision
6	mAP	mean Average Precision	15	R	Recall
7	GFLOPs	number of floating point operations	16	FPS	Frames Per Second
8	YOLO	You Only Look Once	17	IoU	Intersection over Union
9	SSD	Single Shot MultiBox Detector	18	DFL	Deep Feature Loss

**Table 2 foods-13-01179-t002:** Results of experiments comparing all attention modules.

Model	GFLOPs	Params/M	FPS	mAP/%	P/%	R/%
YOLOv8n	8.858	3.157	166.28	85.49	89.04	82.37
YOLOv8n + CBAM	8.861	3.179	157.63	86.08	90.37	83.36
YOLOv8n + CA	8.861	3.169	151.58	86.23	89.88	82.79
YOLOv8n + ECA	8.859	3.157	164.59	86.14	88.95	82.89
YOLOv8n + NAM	8.863	3.158	162.40	86.17	89.49	83.28

Note: GFLOPs represents the number of floating point operations performed; Params is the number of model parameters; FPS is the speed of model inference; mAP is the average of the detection accuracy; P stands for precision; and R stands for recall.

**Table 3 foods-13-01179-t003:** Results of ablation experiments.

Model	GFLOPs	Params/M	FPS	mAP/%	P/%	R/%
YOLOv8n	8.858	3.157	166.28	85.49	89.04	82.37
YOLOv8n + CBAM	8.861	3.179	157.63	86.08	90.37	83.36
YOLOv8n + C2fGhost	7.975	2.725	175.39	86.42	89.47	82.84
YOLOv8n + CBAM + C2fGhost	7.978	2.746	159.94	86.69	89.70	82.81
YOLOv8n + CBAM + C2fGhost + SIoU	7.978	2.746	165.80	86.16	89.77	83.01

Note: GFLOPs represents the number of floating point operations performed; Params is the number of model parameters; FPS is the speed of model inference; mAP is the average of the detection accuracy; P stands for precision; and R stands for recall.

**Table 4 foods-13-01179-t004:** Experimental results for comparisons with other advanced models.

Model	GFLOPs	Params/M	FPS	mAP50/%	P/%	R/%
Faster R-CNN	370.210	137.099	22.90	87.07	76.69	86.19
SSD	62.747	26.285	132.23	87.71	82.65	84.55
YOLOv3	66.171	61.949	96.95	85.47	87.91	80.93
YOLOv4_Tiny	6.957	6.057	242.49	83.45	84.07	80.05
YOLOv5n	4.564	1.872	156.99	85.87	88.73	81.66
YOLOv7	106.472	37.620	57.25	88.17	90.76	84.74
YOLOv8n	8.858	3.157	166.28	85.49	89.04	82.37
YOLOv8n + CBAM + C2fGhost + SIoU	7.978	2.746	165.80	86.16	89.77	83.01

Note: GFLOPs represents the number of floating point operations performed; Params is the number of model parameters; FPS is the speed of model inference; mAP is the average of the detection accuracy; P stands for precision; and R stands for recall.

## Data Availability

The original contributions presented in the study are included in the article, further inquiries can be directed to the corresponding authors.

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
