# Peer review of "Quality Detection and Grading of Rose Tea Based on a Lightweight Model"

_foods, 2024, doi:10.3390/foods13081179_

Round 1

Reviewer 1 Report

Comments and Suggestions for Authors

This study shows the potential application of a lightweight model for quality detection and grading of rose tea. The results are promising and interesting. However, I made some comments to clarify some issues.

1. Line 13. Please use consistent terms. Lightweight model or lightweight algorithm?

2. Please add sample information such as seven types of rose tea in the abstract.

3. Line 31. The reference should be in the numbering system. Please follow the journal guidelines.

4. In the introduction, the reason why using a lightweight model is well presented. However, the reason for comparing several different algorithms is not clear.  

5. The title of the figures is not well presented. For example, Figure 6. NAM. It is not easy to understand the meaning. Please provide an easy and understandable title.

6. Line 215. Please provide GPS information for the rose tea plantation used in this study.

7. Line 225. There is no information about the definition of waste flower.

8. Line 228. Why there is no waste of flower images?

9. Line 232. Why were two cameras used? for the calculation which camera was used? Canon camera (please provide a model in detail) or mobile phone? What is the difference between the two?

10. Line 234. How the partition of the samples into training and validation was conducted? random? or using some algorithm such as the Kennard Stone algorithm?

Reviewer 2 Report

Comments and Suggestions for Authors

The manuscript presents a significant advancement in the field of rose tea grading through the development of a lightweight algorithm for automation. The authors successfully address the inefficiencies of manual grading by introducing innovative techniques within the YOLOv8n network framework. By incorporating attention mechanisms and lightweight modules, namely the Convolutional Block Attention Module (CBAM) and C2fGhost, the authors effectively enhance detection performance while reducing computational complexity.

Moreover, the adoption of the SIoU Loss function demonstrates a thoughtful approach towards improving boundary regression performance. The experimental results showcase notable enhancements in various performance metrics, including mean Average Precision (mAP), Precision (P), and Recall (R), indicating the effectiveness of the proposed methodology. Notably, the improved model outperforms existing detection models, highlighting its significance in practical applications for rose tea grading.

While the manuscript presents a comprehensive approach to automating rose tea grading, there are several areas where improvements could enhance the clarity and impact of the research:

1.      Addressing the limitations of the proposed algorithm and potential challenges in real-world implementation would provide a more balanced perspective.

2.      The discussion section in the manuscript appears relatively scarce and could benefit from expansion to provide deeper insights into the implications of the findings. A more robust discussion would involve analyzing the significance of the results within the broader context of rose tea grading and automation. This could include considerations of potential applications and challenges.

3.      Similarly, the conclusion and future outlook section could be further elaborated to highlight the key contributions of the study and outline specific avenues for future investigation. By enhancing these sections, the manuscript would offer a more comprehensive understanding of the proposed algorithm's relevance and potential impact in the field of rose tea grading.

4.      It is recommended to rework certain fragment of the manuscript for clarity and precision. For instance, consider revising some sentences, for example: “Therefore, choosing YOLOv8 model improvement is correct” or “It offers technical assistance and theoretical study for rose tea grading that is clever”.

Comments on the Quality of English Language

4.      It is recommended to rework certain fragment of the manuscript for clarity and precision. For instance, consider revising some sentences, for example: “Therefore, choosing YOLOv8 model improvement is correct” or “It offers technical assistance and theoretical study for rose tea grading that is clever”.

Reviewer 3 Report

Comments and Suggestions for Authors

I consider the description of the algorithms to be a very thorough job. Nevertheless, in my opinion this manuscript does not belong in this journal, but in a chemometrics journal.

Round 2

Reviewer 3 Report

Comments and Suggestions for Authors

I remain of the opinion that this manuscript is more suited to a chemometric journal.

If the editor has decided that Foods will accept the manuscript, then I accept the addition to my comment 2. I recommend it for publication.